# Flexible and Efficient Long-Range Planning Through Curious Exploration

## Abstract

Identifying algorithms that flexibly and efficiently discover temporally-extended multi-phase plans is an essential next step for the advancement of robotics and model-based reinforcement learning. The core problem of long-range planning is finding an efficient way to search through the tree of possible action sequences — which, if left unchecked, grows exponentially with the length of the plan. Existing non-learned planning solutions from the Task and Motion Planning (TAMP) literature rely on the existence of logical descriptions for the effects and preconditions for actions. This constraint allows TAMP methods to efficiently reduce the tree search problem but limits their ability to generalize to unseen and complex physical environments. In contrast, deep reinforcement learning (DRL) methods use flexible neural-network-based function approximators to discover policies that generalize naturally to unseen circumstances. However, DRL methods have had trouble dealing with the very sparse reward landscapes inherent to long-range multi-step planning situations. Here, we propose the Curious Sample Planner (CSP), which fuses elements of TAMP and DRL by using a curiosity-guided sampling strategy to learn to efficiently explore the tree of action effects. We show that CSP can efficiently discover interesting and complex temporally-extended plans for solving a wide range of physically realistic 3D tasks. In contrast, standard DRL and random sampling methods often fail to solve these tasks at all or do so only with a huge and highly variable number of training samples. We explore the use of a variety of curiosity metrics with CSP and analyze the types of solutions that CSP discovers. Finally, we show that CSP supports task transfer so that the exploration policies learned during experience with one task can help improve efficiency on related tasks.

## 1 Introduction

Many complex behaviors such as cleaning a kitchen, organizing a drawer, or cooking a meal require plans that are a combination of low-level geometric manipulation and high-level action sequencing. For example, boiling water requires sequencing high-level actions such as fetching a pot, pouring water into the pot, and turning on the stove. In turn, each of these high-level steps consists of many low-level task-specific geometric action primitives. For instance, grabbing a pot requires intricate motor manipulation and physical considerations such as friction, force, etc. The process of combining these low-level geometric decisions and high-level action sequences is often referred to as multi-step planning.

While high-level task planning and low-level geometric planning are difficult problems on their own, integrating them presents unique challenges that add further complexity. Task and Motion Planning (TAMP) is a powerful approach to the problem which constructs plans in logical terms that execute a sequence of macro-actions that are composed of geometric motion plans (Fikes & Nilsson, 1971; Dantam et al., 2016). While TAMP has been successful at generating temporally extended multi-step plans that conform to geometric constraints, it requires actions to have defined preconditions and effects that modify the logical description of the world state. This is an unreasonable assumption for complex physical tasks because real-world effects and preconditions are often unknown or difficult to describe with logical predicates. These assumptions limit the flexibility and robustness of the TAMP approach. In addition, TAMP is computationally costly because it requires geometric motion planning for each macro-action sample. In contrast, deep reinforcement learning (DRL) methods

have shown success at learning flexible policies for a variety of complex tasks in unstructured domains (Arulkumaran et al., 2017). However, DRL methods typically require very large sample sizes of successful trajectories to learn useful policies and thus have trouble in sparse reward landscapes (Choi et al., 2019; Riedmiller et al., 2018). Multi-step planning, by its very nature, involves sparse reward landscapes, as highly specific action sequences must be executed before any positive rewards are observed. As such, flexible multi-step planning is an open challenge in artificial intelligence and algorithmic robotics.

In this paper we combine aspects of TAMP and DRL to achieve progress toward efficient and flexible multi-step planning. We introduce the Curious Sample Planner (CSP), which uses curiosity to bias the search through action space toward novel points in the state space. Our method combines the flexibility and transferability of DRL with the temporally extended multi-step planning of TAMP. We illustrate the power of CSP in a variety of qualitatively distinct problems in multi-step planning for which logical descriptions of action effects would be difficult to construct, including the building of complex structures and the discovery of simple machines for achieving challenging physical goals. We show CSP dramatically improves sample complexity compared to random TAMP exploration and DRL baselines. We compare a variety of distinct curiosity metrics for use with CSP and demonstrate that dynamics-based curiosity performs well on tasks which require many dynamic object interactions while state-based curiosity performs better for structure-building tasks. Finally, we show that CSP can be used to achieve substantial transfer between related but qualitatively distinct tasks, utilizing knowledge in one domain to increase performance in a similar domain.

## 2 RELATED WORK

Finding policies in sparse-reward environments has been an ever-present challenge for modern deep reinforcement learning. Model-free RL algorithms either require millions of samples in a sparse-reward setting (Mnih et al., 2015; Hessel et al., 2017) or a precise reward curriculum that gives checkpoints or gradients toward the goal (Heess et al., 2017). While these algorithms yield convincing performance, humans can learn these tasks without strict reward curricula or massive sample sizes.

Model-based RL presents a promising approach for solving these sparse-reward tasks by allowing agents to create plans through virtual experimentation. Rather than learning a direct policy or value function, model-based RL focuses on learning the local or global dynamics of the task to enable planning (Kaiser et al., 2019; Levine et al., 2015) or to create imagined environment roll-outs for more efficient policy training (Weber et al., 2017; Ha & Schmidhuber, 2018). Once the structure of the task is known, the question becomes how to best plan under that structure. Our algorithm is a method for multi-step planning after the environmental dynamics and task structure are known.

A classical approach to planning is that of sample-based geometric motion planners. Rapidly Exploring Random Trees (RRT) and Probabilistic Road Maps (PRM) combine goal-directed sampling with off-target sampling to balance exploration of the state space with exploitation of knowledge about the goal configuration (Lavalle, 1998; Kavraki et al., 1996). While these algorithms can work even for high dimensional configuration spaces, they are computationally intractable for tasks with the complex constraints that often exist in real world settings (Kingston et al., 2018). For a robotic manipulator, grasping an object has a necessary condition that the agent's manipulator is in a position to grasp that object. These constraint barriers in the configuration space render the goal-directed component of motion planning ineffective and necessitate a random exploration the entire configuration space. So while sample-based geometric motion planners are effective at simple tasks, they fail for temporally extended multi-step tasks with intricate constraints. We use sample-based geometric motion planners as a part of our solution to the multi-step planning problem.

More recently, Task and Motion Planning has shown success in developing temporally extended multi-step plans under both fully known and partially observed environment dynamics. A number of TAMP algorithms (Kaelbling & Lozano-Pérez, 2013; Gravot et al., 2005; Hertle et al., 2012; Srivastava et al., 2014) have solved tasks such as block stacking, object packing, table setting, and much more. TAMP algorithms generally iterate between a motion planning step (using RRT, PRM, etc) and a symbolic planning step using task planning algorithms such as Fast Downward Planning (Helmert, 2006) or Fast Forward Planning (Hoffmann & Nebel, 2001). However, because geometric motion planning is relatively slow, Task and Motion Planning is computationally burdensome. Ad-

ditionally, defining logical predicates for the effects and preconditions of macro-actions is a manual undertaking which limits the flexibility of the solution. We take inspiration from Task and Motion planning by using geometric motion planners as procedures used in executing macro-actions, but increase planning efficiency and flexibility by intelligently sampling macro-actions and avoiding logical descriptions of effects and preconditions.

Many other methods have been proposed that increase the efficiency and flexibility of Task and Motion Planning. Supervised learning of the preconditions and effects of certain macro-actions removes the need for manual specification, but requires constructing task-specific training datasets (Kroemer & Sukhatme, 2016; Wang et al., 2018). Other work has focused on factoring the planning problem into submanifolds with analytic constraints in order to reduce the size of the search space. (Garrett et al., 2018; Vega-Brown & Roy, 2018). TAMP can be accelerated by generating a sampling distribution around the goal trajectory using GANs (Kim et al., 2018) or by using reinforcement learning to learn search heuristics through expert examples or previously solved problem instances (Chitnis et al., 2016; Kim et al., 2019). In this work, we also aim to build an algorithm that can utilize information gained from previous problem instances to speed up planning and generalize to related tasks. However, instead of supervising on expert examples or task-specific training sets, we aim to use a self-supervision signal to increase the efficiency of multi-step planning.

Along with model-based RL and planning, intrinsic motivation, also known as curiosity, has been used to turn sparse reward spaces into dense ones by self-supervising on an auxiliary task. Some commonly used auxiliary tasks are forward dynamics (Burda et al., 2019), random network distillation (Choi et al., 2019), and state estimation (Mitash et al., 2017). The error from these predictor networks are considered an intrinsic reward for some reward optimization algorithm. These auxiliary tasks encourage state-space and action-space coverage and increase the probability of encountering sparse rewards over random policies with epsilon-greedy exploration (Jaderberg et al., 2016). Our algorithm uses the principle of self-supervision on auxiliary tasks to accelerate goal discovery in multi-step planning.

## 3 MATHEMATICAL DESCRIPTION

### 3.1 PROBLEM DESCRIPTION

In its general formulation, our problem consists of a robot $\mathcal{R}$ in an environment with dynamic movable objects $D_1, D_2, ...D_k$ and static objects $S_1, S_2, ...S_l$. The robot $\mathcal{R}$ has an associated configuration space $\mathcal{Q}$ containing its possible configurations in its environment, so that $dim(\mathcal{Q})$ is the number of degrees of freedom of $\mathcal{R}$. There is a larger state-space $\mathcal{S}$ which, in addition to $\mathcal{Q}$, includes includes the total description of the positions and orientations of the dynamic and static objects, as well as their velocities, masses, and shapes, together with any (possibly dynamic) state-space constraints such as links or joints that may (permanently or temporarily) exist between objects in the environment.

The robot is equipped with a space $\mathcal{A}$ of primitive actions that cause reliable (though possibly stochastic) changes in the state. The robot is also equipped with a set of macro-actions $\mathcal{M}$ that can be translated into sequences of motion primitives in $\mathcal{A}$ using geometric motion planning and inverse kinematic procedures. Each macro-action is parameterized by specific continuous and discrete parameters, and has a set of implicitly-defined feasible conditions.

In our specific experiments, we use a mounted robot arm with seven bounded revolute joints resulting in a seven dimensional configuration space, with fully-extended length $r$. Our robot's action primitives $\mathcal{A}$ consist of (1) a 7-dimensional continuous rotation space composed by linearly interpolating between $\theta_1$ and $\theta_2$ where $\theta_1, \theta_2$ are 7-vectors corresponding to bounds on each rotational degree of freedom, and (2) the ability to add a link constraint between any two objects in the environment, as long as the objects are touching and the robot arm is sufficiently close to both. We use Bullet (Coumans, 2015), a flexible physics simulation library, to execute robot actions in simulation. The robot has access to a `PickPlace` macro-action which takes a target object, a goal position, and a goal orientation as input, and moves the target block to the goal position and orientation. The robot also has access to `AddConstraint` macro which takes two target objects as input, moves them into position for linking, and performs the link; and a corresponding `RemoveConstraint` that breaks the link if the objects are connected. While this setup is fairly typical of robotics appli-

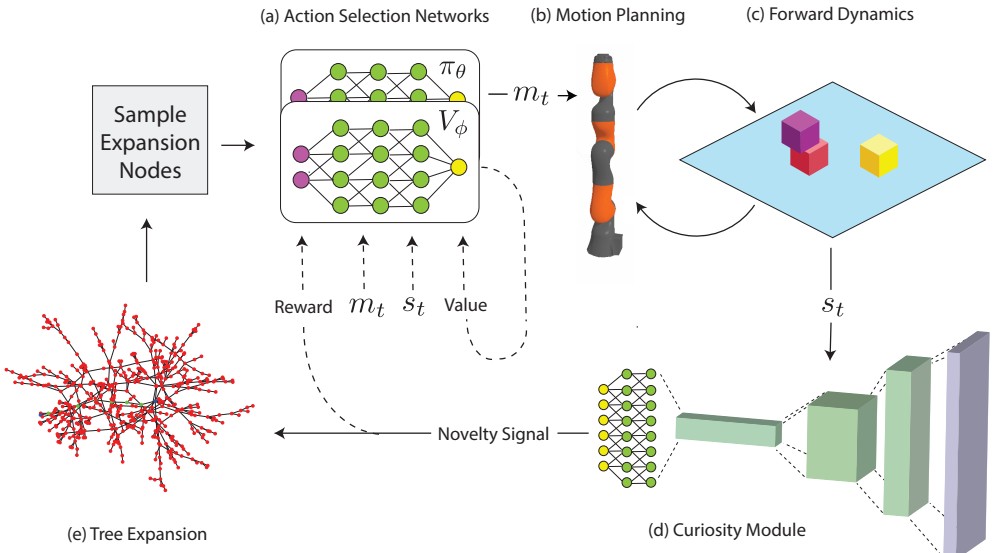

**Figure 1:** a.) The action selection networks select actions that maximize curiosity. b.) Parameterized macro-actions are converted to motion primitives c.) The forward dynamics module predicts the effects of executing those motor primitives in a particular state. d.) The curiosity module represents the current world state and outputs one of the curiosity metrics (see section 4.3). e.) The search tree is expanded.

cations, the CSP algorithm we describe below makes no assumptions about the effects of actions on objects in the environment, and so our method could be applied to more complicated environments containing soft-body objects, cloths, and multiple robotic manipulators.

The robot must use its actions and macro-actions to achieve a goal, defined by a state-space subset $G \subseteq \mathcal{S}$, by manipulating the state from some specified initial conditions into the $G$ subset. For example, for our robotic arm, the task of moving a dynamic object $D_i$ outside the reachable radius $r$ of the robot would be defined by $G = \left\{ s \in \mathcal{S} | D_{i_x}{}^2 + D_{i_y}{}^2 > r^2 \right\}$. Though this type of goal definition is very simple and natural, finding a solution to such a task require complex strategic planning — e.g. perhaps requiring the discovery and construction of a simple machine, such as a ramp that could be used to slide the block out of the robot's reach (see section 4.1).

Finally, the robot has access to a (deterministic) dynamics model, $f : \mathcal{S} \times \mathcal{A} \to \mathcal{S}$, that maps states and actions to future states i.e. $f(s_t, a) = s_{t+\tau}$ for some $\tau > 0$. In this work, $f$ simply gives the robot access directly to the underlying Bullet physics simulator as a black box, but future research could involve having $f$ be a learned (possibly non-deterministic) forward dynamic prediction network that must be inferred from agent experience (such as e.g. (Mrowca et al., 2018)).

### 3.2 CURIOUS SAMPLE PLANNER

Below, we describe the Curious Sample Planner. We first describe the CSP's system architecture, including its constituent neural networks and geometric planning module. We then describe the core curious tree-search algorithm by which CSP uses these modules to construct multi-step plans.

**System Architecture:** CSP is comprised of four main modules (Fig. 1). The action selection networks include an actor-network $\pi_\theta : \mathcal{S} \to \mathcal{M}$ and a critic-network $V_\phi : \mathcal{S} \to \mathbb{R}$ (Fig. 1a), which learn to select macro-actions and choose parameters of that macro-action given a particular state. The action selection networks have two primary functions: maximizing curiosity in action selection and avoiding infeasible macro-actions. The networks are trained using actor-critic reinforcement learning (PPO (Schulman et al., 2017)) where $\pi_\theta$ has learnable parameters $\theta$ and $V_\phi$ has learnable parameters $\phi$. The networks select feasible actions which maximize the novelty signal,

leading to actions which result in novel configurations or dynamics. The actor network outputs a continuous (real-valued) vector which is translated into a macro-action with both discrete and continuous parameters. The forward dynamics module $f : \mathcal{S} \times \mathcal{A} \rightarrow \mathcal{S}$ (Fig. 1c) takes a state and an action primitive, simulates forward a fixed time $\tau$, and returns the resulting state. This forward dynamics module is used by a geometric planning module (Fig. 1b) to convert macro-actions in $\mathcal{M}$ into feasible sequences of motor primitives in $\mathcal{A}$. Finally, the curiosity module is a neural network $H_\beta$ ($H$ stands for heuristic, Fig. 1d) that takes states as inputs and returns a curiosity score, with learnable parameters $\beta$. The exact input and output of the curiosity module is dependent on the type of curiosity being used (see §4.3).

---

**Input:** Initial state $s_0$, Goal set $G$, dynamics $f$
**Output:** Path$\{(s_0, a_0, s_1), ..., (s_{n-1}, a_{n-1}, s_n)\}$ where $s_n \in G, f(s_i, a_i) = s_{i+1}$

---

1   $\mathcal{T} = (\{s_0\}, \emptyset)$
2   Randomly Initialize $\pi_\theta, V_\phi, H_\beta$
3   Initialize P$(s_0) = 1$
4   **while** $\mathcal{V} \cap G = \emptyset$ **do**
5     $S \leftarrow$ batch sample $\sim$ P$(\mathcal{V})$;                     `// Expand novel states`
6     $M \leftarrow \pi_\theta(S)$
7     $A =$ motion planning using $f$ for each $m \in M$
8     $S' \leftarrow f(S, A)$
9     $\mathcal{L} \leftarrow$ novelty metric ;         `// Various metrics described in sec 4.3`
10    $\mathcal{L}(\mathcal{A} = \emptyset) = 0$ ;            `// No reward for infeasible actions`
11    Train $H_\beta$ to minimize $\mathcal{L}(\mathcal{A} \neq \emptyset)$
12    Update $\pi_\theta, V_\phi$ to maximize $\mathcal{L}$;               `// PPO Training`
13    $\mathcal{V} \leftarrow S' \cup \mathcal{V}$
14    Add each $(S_i, S'_i)$ to $\mathcal{E}$
15    P $=$ softmax(novelty metric for each $v \in \mathcal{V}$)
    **end**
16  **return** *Path in $\mathcal{T}$ from $s_0$ to $s_g \in \mathcal{V} \cap G$*

**Algorithm 1:** The CSP algorithm.

**CSP Algorithm:** At its core, CSP is an algorithm for efficiently building a search tree over the state space using parameterized macro-actions (see Algorithm 1). The algorithm starts by initializing a tree $\mathcal{T} = (\mathcal{V}, \mathcal{E})$ in which the vertices are points in the continuous state space, edges are macro-actions with defined parameters, and paths are sequences of macro-actions which are guaranteed to transition between states at each end of the path under the dynamics model. The tree starts with a single vertex $s_0$ which is the start state of the multi-step planning problem. The algorithm also initializes a probability distribution over the tree vertices such that $P(s_0) = 1$. A batch of size $B$ states are then sampled from $P$ and passed into $\pi_\theta$ resulting in a set of state/macro-action pairings.

The next step of the algorithm is to convert the selected macro-actions into primitive action sequences using a combination of inverse kinematics and geometric motion planning. We use RRT-Connect (Kuffner & LaValle, 2000) for motion planning and the recursive Newton Euler algorithm (Luh et al., 1980) for inverse kinematics. The exact routine for converting macro-actions to primitives is specific to the macro-action. In some cases it is infeasible to convert macro-actions to motor primitives. (For example, it is infeasible to pick up an object that is out of the robot's reach.) In such cases, the planning module returns an empty sequence of primitives. These feasibility conditions are not explicitly represented as logical preconditions, but are discovered from failed attempts at inverse kinematics or motion planning. Over time, the action selection network learns to focus only on feasible macro-actions. The resulting sequence of action primitives are passed into the black-box dynamics module to get a corresponding batch of future states.

In order to determine which states and actions should be further explored, the algorithm creates a curiosity score for each of the selected macro-actions. Passing the batch of states and future states through the curiosity module will give a prediction loss, or novelty score. A subset of the new states with high novelty scores in the batch are then added as vertices in the search tree and the probability distribution $P$ is adjusted to give more weight to states with high novelty scores. Although it is not strictly necessary for CSP to function, for increased computational efficiency we discard vertices

$(a)$ STACK   $(b)$ PUSH-AWAY   $(c)$ BOOKSHELF   $(d)$ LAUNCH-BLOCK

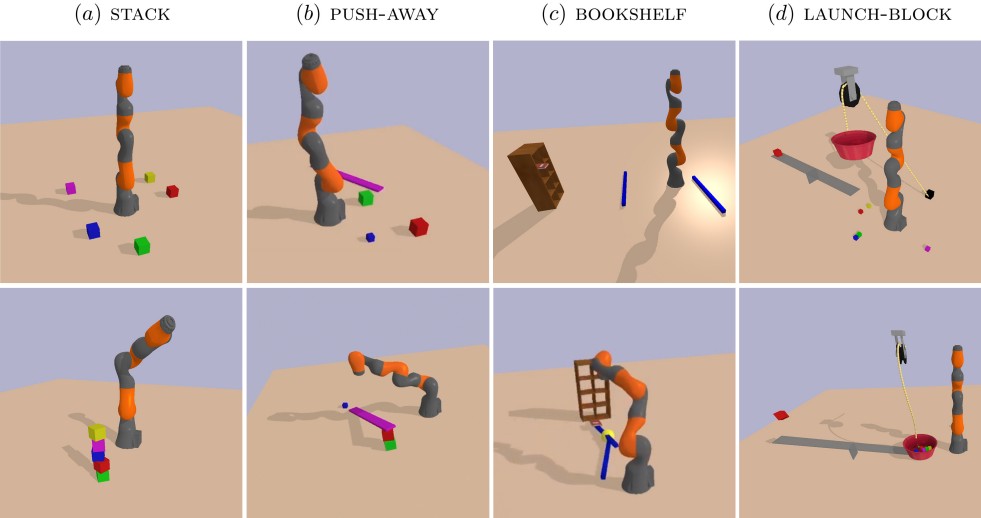

**Figure 2:** Visualizations each of the four task categories used to test CSP. Top Row: representative initial state for each task. Bottom Row: representative final state for each task. The **Block-Stack** task requires the robot to find a way to cause one block to remain stably at a high $z$-position without touching the arm, necessitating the building of a tower. The **Push-Away** task requires the robot to push the small (blue) block beyond the reach of the robot arm, which, depending on the circumstance might require the robot to build a ramp. The **Bookshelf** tasks requires the robot to knock a book of a shelf which is further away from the robot that it can reach, necessitating that the robot discover how to build a simple tool from the provided blue rods. The **Launch-Block** task requires the robot to launch the small red block off the far end of the seesaw, but it can only do so (we think) by putting all five of the other blocks as counterweight into the red bucket and detaching the pulley rope, causing the now-heavy bucket to crash into the close end of the seesaw. Tasks are described in more detail in Section 4.1.

with low probability (typically the bottom 90% of the distribution), as they are extremely unlikely to be sampled. After scores are calculated, the input states that have infeasible macro-actions are given a curiosity score of zero, strongly disincentivizing infeasible macro-action selection. The curiosity module is trained from the losses generated by the batch and the novelty score is used as the reward for the training of the action selection networks. This process repeats until the algorithm reaches a state in the goal subset or is exogenously terminated.

# 4 EXPERIMENTS

## 4.1 TASKS

Each of the four task types below are extremely easy to specify in the non-restrictive goal semantics accepted by CSP, but often require complex multi-step planning to solve, and would be extremely statistically improbable to be solved through random exploration. From a DRL point of view, such tasks correspond to extremely sparse reward landscapes. They also involve complex and fairly precise continuous motor manipulation of both rigid and non-rigid objects, using sequences of macro-actions whose preconditions would be very challenging to specify using logical predicates. The task set also affords some natural opportunities for cross-task transfer (e.g. between tower-building in **Block-Stack** and ramp building for **Push-Away**).

**Block-Stack**: In this task, the robot is provided with a set of $K$ cubic blocks of size $h$, and the robot's goal is to cause at least one block to stably remain in position at $z$-position of greater than $(K-1)*h$, without being in contact with the robot arm, for at least two seconds. Of course, the

only feasible way for the robot to solve this problem is to stack the $K$ blocks in a stable tower, but the robot is not provided with any reward at intermediate unsolved conditions. This type of block stacking is a commonly used task for evaluating planning algorithms because it requires detailed motor control and has a goal that is sparse in the state space.

**Push-Away**: In this task, the robot is provided with several large cubic blocks, a flat elongated block, and one additional smaller "target" object that can be of variable shape (e.g. a cube or a sphere) and material (with e.g. high or low coefficient of friction). The objective is to somehow push the target object outside the directly reachable radius of the robotic arm. Depending on the situation, the solution to the task can be very simple or quite complex. For example, if the target object is spherical with low friction, the robot could solve the task simply by dropping it against one of the larger blocks, causing it to roll away. However, if the target object is cubic with high friction, it may be necessary for the robot to discover how to construct and use a simple machine such as a ramp — e.g. consisting of the large blocks stacked up, with the elongated block placed at one end on the stack as an inclined plane down which the small object can be slid.

**Bookshelf**: In this task, the environment contains a bookshelf with a single book on it. The robot is also provided with two elongated rectangular-prism rods initial placed at random (reachable) locations in the environment. The goal is to knock the book off the bookshelf. However, the book and bookshelf are not only outside the reachable radius of the arm, but they are further than the combined length of the arm and a single rod. However, the robot can solve the task by (e.g.) combining the two rods in an end-to-end configuration using the link macro-action, and then using the combined object to dislodge the book.

**Launch-Block**: In this task, the environment contains a pre-built rope-and-pulley with one end of the rope connected to an anchor block on the floor and the other attached to a bucket that is suspended in mid-air. A seesaw balances evenly below the bucket, with a target block on the far end of the seesaw. The goal is to launch this target block into the air, above the seesaw surface. The robot could solve this task by (e.g.) gathering all blocks into the bucket and untying the anchor knot so that the bucket will descend onto the near end of the seesaw. However, due to the masses of the blocks and the friction in the seesaw, this can only happen when the combined weight of all five blocks are used.

## 4.2 Neural Network Settings and Curiosity Metrics

Throughout our experiments, the action selection networks $\pi_\theta$ and $V_\phi$ are three-layer networks with 64 hidden units each, using the $\tanh$ activation function.

There is a wide range of potential curiosity and novelty metrics, and the optimal metric may be task-dependent. For this reason, we explored a range of such metrics from the recent literature, including: State Estimation (SE) (Mitash et al., 2017), Forward Dynamics (FD) (Burda et al., 2019), and Random Network Distillation (RND) (Choi et al., 2019). The neural network architecture of the curiosity module naturally needed to vary as a function of which metric was chosen. For SE, the curiosity module accepted images of the scene generated from multiple perspectives, in the form of $84 \times 84 \times (3 \cdot n_p)$ pixel arrays (where $n_p$ is the number of perspectives taken), while the architecture was a five-layer convolutional neural network with 3 convolution layers and two fully connected layers. For both FD and RND the architecture consisted of a four-layer MLP with 128 units in each layer. For FD, the inputs were the concatenated vector of system states and actions, while for RND the input simply consisted of the state vector. To enable fair comparison, we ensured that the number of trainable parameters in the curiosity modules was approximately even across all curiosity variants.

## 4.3 Quantitative Results

For each task, we tested CSP in comparison to a variety of baseline approaches. Simple random exploration was tested by assigning a uniform curiosity score to each state and selecting macro-actions from a uniform distribution (CSP-No Curiosity). We also tested against vanilla implementations of vanilla PPO (Schulman et al., 2017) and A2C (Mnih et al., 2016) RL algorithms, using the same architecture as the curious action selector shown in figure 1, but without the tree expansion and sampling module. Finally, we tested a non-planning based but still curiosity-driven DRL model, using

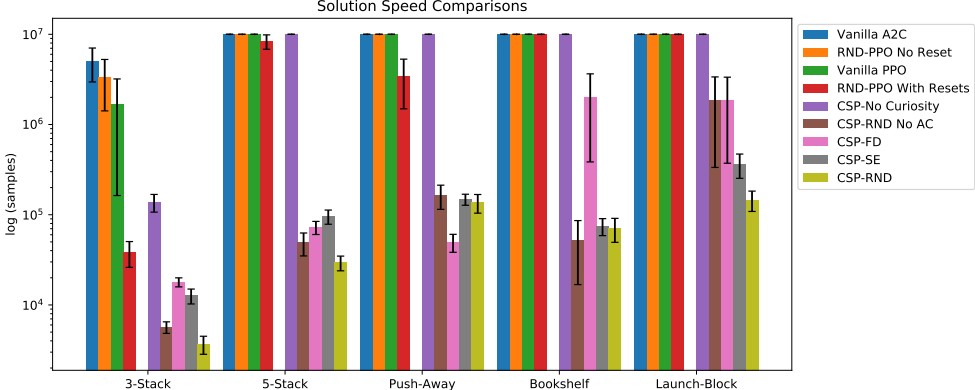

**Figure 3:** Solution speed for all tested algorithm variants across the five tested tasks. Within each group of 7 bars, the left-most bars represent baselines, while the right-most are the CSP models, including the control No-Curosity control and several CSP variants with different curiosity metrics (see text for details). The $y$-axis is on a log scale because the differences between CSP and non-CSP solutions is so great that more subtle differences between various curiosity-metric variants of CSP would otherwise be too small to see. Each bar is computed as the mean of six independent trial runs, and error bars represent the standard error of the mean across trials. We set a maximum number of samples ($10^7$) in order to ensure termination; in many case the baseline algorithms failed to terminate with a successful solution on any trial.

the PPO algorithm with the RND curiosity signal (PPO-RND). All metrics and baselines are compared by measuring the number of macro-action samples needed to reach the goal over six different problem instances on five distinct tasks.

Overall, we found that CSP was dramatically more effective than the baselines at solving all four task types (Fig. 3). The control algorithms (including the CSP-No Curiosity baseline) were sometimes able to discover solution in the simplest cases (e.g. the 3-Block task). However, they were substantially more variable in terms of the number of samples needed; and they largely failed to achieve solutions in more complex cases within a $10^7$-step maximum sample limit. The failure of the random and vanilla PPO/A2C baseliness is not entirely surprising: the tasks we chose here are representative of the kind of long-range planning problems that typically are extremely difficult for standard DRL approaches. The more sophisticated curiosity driven PPO-RND algorithm was able to make headway in a few of the more complex circumstances, but was nonetheless substantially less powerful than the CSP variants.[1] We ran many additional actor critic and curiosity metric combinations and discussed the results in Section D. Comparing between curiosity metrics, we found that, while there is some task dependence, Random Network Distillation serves as a good general-purpose curiosity metric achieving the best performance on four of the five tasks (see figure 3). We also performed an ablation on the action selection networks (CSP-RND No AC) to see how beneficial they were to the planning process over random action selection. We found that in all tasks except Bookshelf, performed slightly better with the action selection networks. While not vital to the planning process, these networks are key for transfer (see sec. 4.5).

---

[1]RL algorithms are often designed with a multi-episodic environment in mind, in which the external controller repeatedly resets the system to a known state (e.g. in a video game level after the time limit has elapsed). Resetting can often help learning algorithms that would otherwise fail, since it rescues them from unrecoverable states. In our case, such an unrecoverable state arises when objects needed to complete the task are prematurely pushed outside the available radius of the robot arm. The CSP algorithm is easily able to handle this circumstance, because it can expand from previous explored nodes in the state space. However, standard methods likely do not have this capacity. To give the baseline algorithms the best chance of working, we thus tested them in two situations: one without trial resetting (the "natural" circumstance for this work), and one with random trial resetting, characterized by a set with probability 1e-4 after each macro-action. Only the PPO-RND algorithm actually benefited from resets, however, and is the only one shown in Fig. 3.

### 4.4 Qualitative Results

In the process of planning, CSP often found interesting state configurations and unexpected problem-solving techniques. A video illustrating some representative solutions found by CSP can be found here: https://youtu.be/7DSW8Dy9ADQ. As an example, we initially developed the Push-Away task with a spherical ball as a target object, hoping CSP would build a ramp and roll the ball out of the reachability zone. However, CSP instead found a simple solution consisting of dropping the ball directly next to another object in order to get enough horizontal velocity to roll out of the reachability zone. In an attempt to avoid this rather trivial solution, we then switched out the ball for a block with a high coefficient of friction. However, again instead of always building a ramp, CSP sometimes made use of its link macro-action to fix the block to one end of the plank and orient the plank so that the block was outsize the reachability zone. However, once the link macro-action was disabled, ramp-building behavior robustly emerged.

### 4.5 Task Transfer

A central aspect of human intelligence is the ability to improve at solving problems over time and to use knowledge gained from solving one problem to more efficiently solve related problems. For this reason, we were curious whether CSP could transfer knowledge between problem instances with different initial conditions, and between different but related problems. We evaluated transfer both the within-task and between-task using 3-Stack, 5-Stack, and Push-Away tasks. We selected these tasks because they all share a common essential skill: stacking blocks.

We tested transfer from each learnable component of the system individually, to identify which components of the system can be effectively reused between problems, including (i) the baseline "no transfer" condition, where neither the curiosity module nor the action selection networks are transferred from another problem instance; (ii) condition where just the curiosity module or action selection networks were transferred; and (iii) a full transfer condition where both the action selection networks and the curiosity module are transferred between tasks.

Results (Fig. 4) show that transferring the curiosity module alone slightly improves planning ability between instances of the same task, but has a minimal effect and can even be detrimental in the case of between-task transfer. On the other hand, transferring the action selection networks leads to a large increase in efficiency in all tasks except for 3-stack. (The 3-stack task is likely unaffected by transfer because it is solved so quickly that the networks have little time to train.) Full transfer results in transfer performance improvements similar to that of transferring the action selection network alone. Gains arose for qualitatively identifiable reasons e.g. once CSP learned to solve the stacking problem, it tries stacking macro-actions much more frequently as an initial guess, which is naturally useful for solving the Push-Away task. Transfer gains are reflected by the fact that the tree search procedure has learned to very effectively reduce the task tree, a particularly dramatic example of which is illustrated in Fig. 4b.

## 5 Conclusion

In this work, we introduce the CSP algorithm, a fusion of classical task-planning and deep RL approaches that is capable of flexible long-range multi-step planning in sparse reward settings. CSP consists of two learnable components which aim to maximize curiosity in action selection and state exploration. The first component is the action selector networks which reduces sampling of infeasible or uninteresting effects in favor of macro-actions and parameters which lead to interesting and unseen effects. The second component is a curiosity module which guides exploration of the state tree toward novel state configurations by increasing the probability that those nodes will be expanded. We show that the addition of these components not only speeds up planning when compared to random action selection and reinforcement learning baselines but also has the ability to make otherwise intractable problems solvable. We also systematically examine the effect of different forms of curiosity on task performance, finding that the RND variant is the most effective overall. Finally, we look at the transferability of solutions to new instances of the same general problem and to related but different problems, showing that networks trained to select actions states which result in interesting configurations perform better than those which are randomly initialized.

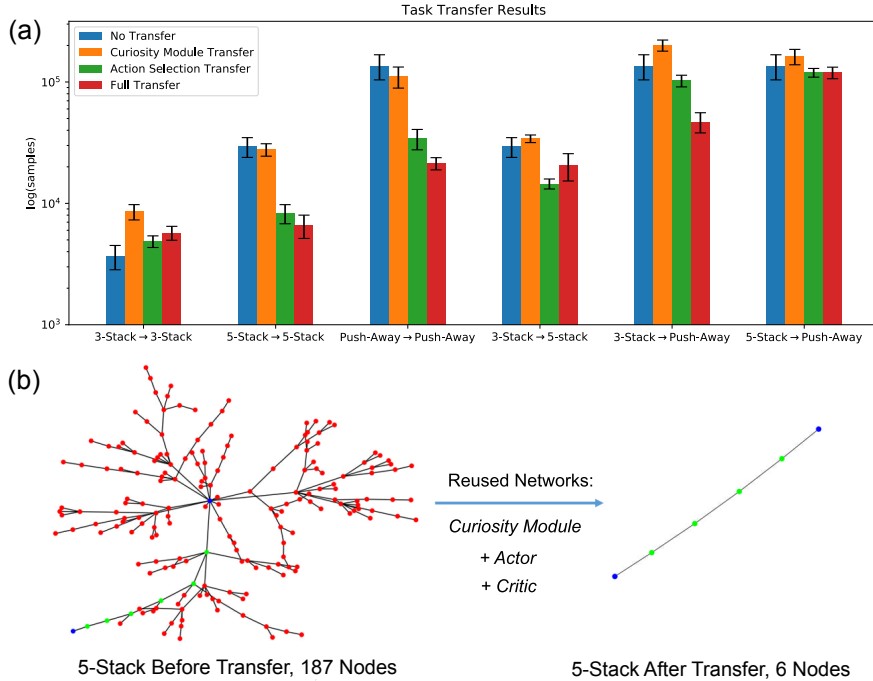

**Figure 4:** (a) Inter-task and between-task transfer efficiency measured by the number of samples needed to reach the goal, for each of several transfer policies (only curiosity module, only action selection module, and full network transfer). The CSP-RND variant is used for all cases. Within each group of bars for a given task, the difference between blue bar (No Transfer) and other bars represents transfer gain (note that the $y$-axis is on a logarithmic scale). Each condition was run 6 times and error bars represent standard error of the mean over runs. (b) Example of search trees for an instance of the 5-stack problem, before (left) and after (right) full network transfer.

Despite its initial successes, the CSP system has a number of key limitations that will need to be addressed in future work. As a pure planning algorithm, CSP does not attempt to solve a directed learning problem over the course of many trials; when very large numbers of trials are available for training, it is plausible that Deep RL approaches would begin to catch up with and eventually exceed CSP performance. A very natural next step will be to combine CSP with a cross-trial learning algorithm to obtain the best of both worlds. It will also be key to show that CSP is able to function effectively when only a non-deterministic and noisy future prediction module is available, rather than the unrealistic perfect black-box predictor used in this work. It will be especially important to determine if CSP is compatible with a learned future predictor, and of interest to determine if CSP-generated plans could help accelerate the learning of such a predictor simultaneously while CSP modules are themselves learned. Another key direction for future work will be to apply a CSP-like procedure to the discovery of macro-actions themselves (Barto et al., 2004; Machado et al., 2017), resulting in a hierarchical multi-timescale procedure in which more and more complex plans become routinized as macro-actions. Finally, it will be of interest to compare the behavior sequences generated by CSP to data from experiments measuring patterns of play and discovery in both human children and adults (Gopnik et al., 2009; Begus et al., 2014), as both the similarities and differences between human decision making and exploration and algorithmic outputs will help inform the development of improved flexible and efficient planning algorithms.

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

## A CURIOSITY METRICS

### A.1 STATE ESTIMATION CURIOSITY

The objective of the state estimation curiosity module is to estimate the underlying state of the world through a set of visual observations. State configurations that are visually distinct from previously seen configurations will lead to high losses. The curiosity module and loss function are defined as follows.

$$H_\beta : Im \mapsto \mathcal{S}$$
$$\mathcal{L}_i = \|H_\beta(Im_i) - s_i\|_2^2$$

Where $Im$ is the space of $84 \times 84 \times 3 \cdot n_p$ matrices containing $n_p$ images captured from various perspectives in the scene, $s \in S$ denotes the state, and $i$ is an index into the batch. In our experiments, we used a single top-down perspective ($n_p = 1$).

### A.2 FORWARD DYNAMICS CURIOSITY

The objective of the forward dynamics curiosity module is to estimate future states given current states and actions. Given deterministic physics, actions modify states deterministically. Therefore, it is theoretically possible to estimate a future state from an action and the current state. Rare or unpredictable object-object interaction will yield the highest loss in dynamics prediction and therefore will be sought out by the curiosity module. The curiosity module and loss function are described as follows.

$$H_\beta : \mathcal{S} \mapsto \mathcal{S}$$
$$f : \mathcal{S} \times \mathcal{A} \mapsto \mathcal{S}$$
$$\mathcal{L}_i = \|H_\beta(s_i) - f(s_i, a_i)\|_2^2$$

Where $f$ is the deterministic dynamics model which maps a state and action at $t$ to a state at time $t + \tau$ for some fixed $\tau$.

### A.3 RANDOM NETWORK DISTILLATION CURIOSITY

Random Network Distillation (RND) is a recently proposed method for obtaining state-space coverage. This is achieved by training a model to fit some fixed random transformation of the input data. The RND curiosity module achieves low loss when it matches the transformation. This results in high curiosity module loss for unseen states.

$$H_\beta, \Phi_\beta : S \mapsto \mathcal{S}$$
$$\mathcal{L}_i = \|H_\beta(s_i) - \Phi(s_i)\|_2^2$$

Where $\Phi$ is the fixed mapping that is initialized with random parameters. Among other mappings including random linear and non-linear mappings with uniformly and gaussian distributed weights, we found that the best performing mapping was a random permutation of the state vector.

## B NETWORK STRUCTURES AND HYPERPARAMETERS

### B.1 ACTOR CRITIC POLICY AND VALUE NETWORKS

The structure of the policy network is a multilayer perceptron model with three layers. The input size is the dimensionality of the state space and the output size is equivalent to the dimensionality of the action space. Action space and state space dimensionality are described in more detail below. The hidden layers each have 64 hidden units with $tanh$ activation functions. The the value network has an equivalent architecture but with an output size of one.

For PPO and A2C training, we used the same hyperparameters during action selection network training and baseline experimentation. For PPO we used a batch size of 128 with 1024 samples collected per update, $\gamma = 0$, $lr = 7e - 4$, $\varepsilon = 0.2$. We chose $\gamma = 0$ for multiple application-dependent reasons. For the curiosity networks, we found that it led to higher performance in both solution speed and selection of feasible actions. In our curiosity-guided RL baselines, we found no difference between values of gamma. We also selected value and entropy coefficients of 0.5 and 0.0 respectively and restricted the magnitude of the gradient to 0.5. For each batch, we did 4 PPO epoch updates. For A2C, we used the same learning rate and coefficients as PPO.

## B.2 STATE ESTIMATION CONVOLUTION NETWORK

The state estimation network takes in visual input from perspectives in the scene and outputs an estimated state. We achieved this image to state mapping using a convolutional neural network with three convolutional layers and two fully connected layers, all with $ReLU$ nonlinearities. Because we used a single perspective for all of our experiments, the input size is $84 \times 84 \times 3$ and the output size is the dimensionality of the state space. The convolutional layers have kernel sizes of 8, 4, 3 and strides of 4, 2, 1. The fully connected layers each have 128 hidden units.

## B.3 FORWARD DYNAMICS NETWORK

The forward dynamics network maps $s_t, a_t$ to $s_{t+\tau}$. This is achieved using a multilayer perceptron model with 3 layers, 64 hidden units per layer and $tanh$ activation functions. The input size is the sum of the action and state dimensionality and the output size is the dimensionality of the state.

## B.4 RND NETWORK

The RND network maps states to transformed states. The size we used for the transformed state is equivalent to the size of the state. Therefore, the input and output sizes are both equivalent to the dimensionality of the state. The architecture consisted of a three layer multilayer perceptron model with $tanh$ activation functions and 64 hidden units for each hidden layer.

## B.5 UNIVERSAL CURIOSITY MODULE HYPERPARAMETERS

For training the curiosity networks, we used an Adam optimizer with a learning rate of $5 \times 10^{-5}$, a batch size of 128, and 1024 samples per update. We also found that performance was higher when we used an adaptive number of samples per update. The number of samples is increased until the loss is below a threshold after training. This improves CSP by giving the curiosity network time to train to baseline before adding nodes to the tree.

## C TASK SPECIFIC STATE SPACE AND ACTION SPACE

### C.1 STACK STATE

This set of tasks consists of k cubes. Each cube has it's own $SE(3)$ configuration parameterized by three positional degrees of freedom and three euclidean rotational degrees of freedom. When the linking macro-action is enabled, the state also contains indicator values for each pairwise object combination.

### C.2 PUSH-AWAY STATE

This task contains two larger cubes, one smaller cube, and one elongated, flat rectangular prism. Again, the dimensionality of the state space is $SE(3)$ for each object along with pairwise indicators when the linking macro-action is enabled.

| Metric-Algorithm | 3-Stack | 5-Stack | Push-Away | Bookshelf | Push-Away |
|---|---|---|---|---|---|
| RND-PPO | $3.34e6 \pm 1.92e6$ | $1e7 \pm 0$ | $1e7 \pm 0$ | $1e7 \pm 0$ | $1e7 \pm 0$ |
| Forward Dynamics-PPO | $5.01e6 \pm 2.03e6$ | $1e7 \pm 0$ | $1e7 \pm 0$ | $1e7 \pm 0$ | $1e7 \pm 0$ |
| Effect Prediction-PPO | $1e7 \pm 0$ | $1e7 \pm 0$ | $1e7 \pm 0$ | $1e7 \pm 0$ | $1e7 \pm 0$ |
| RND-A2C | $2.72e6 \pm 1.92e6$ | $1e7 \pm 0$ | $1e7 \pm 0$ | $1e7 \pm 0$ | $1e7 \pm 0$ |
| Forward Dynamics-A2C | $1e7 \pm 0$ | $1e7 \pm 0$ | $1e7 \pm 0$ | $1e7 \pm 0$ | $1e7 \pm 0$ |
| Effect Prediction-A2C | $1e7 \pm 0$ | $1e7 \pm 0$ | $1e7 \pm 0$ | $1e7 \pm 0$ | $1e7 \pm 0$ |

**Table 1:** Comprehensive List of Actor-Critic/Curiosity Metric Evaluations

## C.3 BOOKSHELF STATE

This task contains two rods elongated rectangular prism rods and a rectangular prism book, each with a $SE(3)$ configuration space and pairwise linking. Linking is always enabled for this task because it is required to complete it.

## C.4 LAUNCH-BLOCK STATE

The state space for this task contains five small cubes, one larger cube that sits at the end of the seesaw, and the angle of the seesaw itself. Each of the cubes has an $SE(3)$ configuration space and the seesaw has a bounded real configuration with a single degree of freedom.

## C.5 TRANSFER STATES

When comparing between tasks, the action spaces and state spaces need matching dimensionality to ensure the networks could be substituted. We achieved this by making sure each problem instance had an equivalent number of objects, even if it was unnecessary to use all objects to complete the task. For the transfer results stated in this paper, this equated to using five objects for each task.

## C.6 ACTION SPACES

When both linking and pick-place macro-actions are enabled, the action space needs to contain information for choosing which object to move ($k$ total), the goal pose of the object ($dof$), which of the $\binom{k}{2}$ object pairs to link or unlink , and which macro-action to select (linking or pick-place). Therefore, the dimensionality of the action space is $\binom{k}{2} + (dof + 1) \cdot k + 2$. The macro-action, link, and object discrete variables are chosen by performing an argmax over the respective section of the action space. If a link macroaction is selected between two objects that are already linked, this is considered an unlink action and the constraint between the two objects is removed. If a link is chosen between two objects which aren't in contact, then the action is considered to be infeasible. For implementation simplicity, we use $k^2$ variables for the linking indicator rather than $\binom{k}{2}$ and ignore any redundant pairs.

## D ADDITIONAL BASELINES

We ran several other reinforcement learning baseline comparisons including Forward Dynamics-PPO, Effect Prediction-PPO, RND-A2C, Forward Dynamics-A2C, and Effect Prediction-A2C without resets, and results are shown in Table 1. As expected, we found results similar to RND-PPO No Reset in Figure 3. Specifically, we found that all additional baselines solved 3-Stack in up to two of the experiments, but failed at ever solving the other tasks.

