# OpenReview forum: "Flexible and Efficient Long-Range Planning Through Curious Exploration"
_ICLR.cc/2020/Conference — Reject_

### Official Review · AnonReviewer3 · 2019-10-23
**Official Blind Review #3**

**Rating:** 3

**Review:**

===== Summary =====
The paper introduces Curious Sample Planner (CSP) a long-horizon motion planning method that combines task and motion planning with deep reinforcement learning in order to solve simulated robotic tasks with sparse rewards. The CSP algorithm considers two different hierarchies of actions: primitive actions, which control the rotation of several joints in a robotic arm, and macro-actions corresponding to complex behaviours  such as moving from one position to another or linking two objects together. Macro-actions are selected using the actor-critic architecture PPO and then turned into primitive actions using geometric motion planning and inverse kinematics. Specifically, RRT-Connect is used for motion planning with the recursive Newton-Euler algorithm for inverse kinematics on a perfect model of the environment to determine the specific sequence of primitive actions necessary to execute the macro-action. As CSP is interacting with the environment, it also builds a tree of states in the environment connected by the macro-actions leading to each of them. Each vertex of the tree is assigned a curiosity score, which is used as an exploration bonus for PPO and to determine the probability with which each vertex is sampled from the tree for future exploration. The whole process is repeated until a feasible path from the initial state to the goal state is found. The paper provides empirical evaluations in four different tasks where it compares the performance of CSP with three different curiosity measures to the performance of PPO and A2C. The results show that CSP accomplishes each task while using significantly less samples. Moreover, a second set of experiments is presented that show the potential for transfer learning across tasks using CSP.

Contributions:
1. The paper introduces CSP, a successful combination of task and motion planning and deep reinforcement learning that can discover temporally extended plans.
2. The paper demonstrates a statistically significant improvement in performance over PPO and A2C in the four robotic tasks that the paper studies.
3. The paper shows evidence that CSP might facilitate transfer learning across similar tasks.

===== Decision =====
The paper represents a significant contribution to the reinforcement learning and task and motion planning literature. The main algorithm is well motivated from previous literature and demonstrate a significant improvement over previously proposed deep reinforcement learning methods. Moreover, the ideas are presented clearly and logically throughout the paper and the empirical evaluations clearly support the claims about the performance of CSP. However, I have concerns about the reproducibility of the results because of the little amount of details provided about the hyper-parameter selection and settings and about the network architectures and loss functions. Thus, I consider that the paper should be rejected, but I am willing to increase my score if my comments are properly addressed.

===== Questions and Comments =====

1. Although the ideas in the paper are presented clearly, the algorithms and methods are presented mostly at a very high level. There are no details about the losses used in lines 11 and 12 of Algorithm 1 and there are no specifications about the network architectures and the hyperparameter settings and selection for each algorithm. This raises two concerns. First, this hinders reproducibility and future work by other authors that might be interested in building upon the ideas presented in the paper; this would also decrease the impact of the paper. Two, it is difficult to determine if the comparisons against A2C and PPO were fair without any information about the hyper-parameter selection. Thus, I consider these details should be included and I would consider increasing my score to accept if this was properly addressed. Specifically, I think the paper should include this:
- The hyper-parameter settings for each different algorithm and an explanation about how they were selected.
- A detailed description of the network architectures used in the experiments.
- A definition for the loss functions used for the policy network, the value network, and the curiosity module.

2. As mentioned in the Decision section above, the paper clearly demonstrates an improvement over previously proposed deep reinforcement learning algorithm. However, there are no comparisons to any previously proposed Task and Motion Planning Methods. What motivated this decision?

3. An alternative to using separate networks for the policy and the value function, it could be possible to use a two-headed network with one head for the policy and another one for the value. What was the reason for using two separate networks over this alternative?

4. Were any other alternatives tested for the activation functions of the network?

5. CSP was tested with three different curiosity and novelty metrics, none of which dominated over all the other ones. However, PPO was only tested with one of the measures and A2C had no curiosity measure added to it. Where there any preliminary results that justified this decision? In terms of computation and time, how difficult would it be to include this in the paper?

6. The paper already hinted at this, but macro-actions could be framed within the option framework from Sutton, Precup, & Singh (1999). This would open up the opportunity to apply some of the already proposed methods for option discovery such as the option-critic architecture from Bacon, Harb, & Precup (2016) or the Laplacian framework for option discovery from Machado, Bellemare, & Bowling (2017), which is cited on the paper. Could the authors provide more comments about this line of future work?

===== References =====
Bacon, P., Harb, J., & Precup, D. (2016). The Option-Critic Architecture. Retrieved 17 October 2019, from https://arxiv.org/abs/1609.05140

Marlos C. Machado, Marc G. Bellemare, and Michael H. Bowling. A laplacian framework for option discovery in reinforcement learning. CoRR, abs/1703.00956, 2017. URL http://arxiv.org/abs/1703.00956.

Sutton, R., Precup, D., & Singh, S. (1999). Between MDPs and semi-MDPs: A framework for temporal abstraction in reinforcement learning. Artificial Intelligence, 112(1-2), 181-211. doi: 10.1016/s0004-3702(99)00052-1

**Experience Assessment:**

I do not know much about this area.

**Review Assessment: Checking Correctness Of Derivations And Theory:**

I carefully checked the derivations and theory.

**Review Assessment: Checking Correctness Of Experiments:**

I carefully checked the experiments.

**Review Assessment: Thoroughness In Paper Reading:**

I read the paper thoroughly.

---

> ### Author Response · Authors · 2019-11-07
> **Initial response to Reviewer 3**
>
> Hi Reviewer 3!
>
> Thanks also for the constructive comments.  See below for inline responses to each issue / question.
>
> "The paper represents a significant contribution ..."
>
> Thanks!
>
>
> "However, I have concerns about the reproducibility of the results ... but I am willing to increase my score if my comments are properly addressed."
>
> Got it. We weren't that sure how much detail to put into the paper, but we see this is a real area for improvement. (Rev 1 also is concerned about this issue.) As we explained to Rev. 1, our plan for addressing this is as follows:
>
> 	1. Revise the main text of the paper to clarify several main important issues.
> 	2. Create a detailed additional supplementary document.
> 	3. Post an anonymous public github repo to which we will commit all the project code.
>
> Our goal is to have this ready for your review by 11/11 or 11/12.
>
> ==> Question: does this plan work for you? Other suggestions welcomed!
>
>
> "... the paper should include [LIST OF SPECIFICS]"
>
> Happy to add these things.  Some we'll put into the main paper and some into the supplement.  When done we'll write a comment pointing you to the edits.
>
>
> "... There are no comparisons to any previously proposed Task and Motion Planning Methods. What motivated this decision?"
>
> Great question.  Will address in a separate post due to character limitation.
>
>
> "... An alternative to using separate networks for the policy and the value function, it could be possible to use a two-headed network with one head for the policy and another one for the value. What was the reason for using two separate networks over this alternative?"
>
> That is indeed an alternative to using separate networks for the policy and value function. The short answer is that we adopted our architecture from a widely used reinforcement learning codebase. https://github.com/ikostrikov/pytorch-a2c-ppo-acktr-gail/blob/master/a2c_ppo_acktr/model.py#L208
>
>
> "... Were any other alternatives tested for the activation functions of the network?"
>
> No, we haven't done this.  Do you think this is very important for us to try?  It's sort of not been a major area of exploration in the literature we've looked at, so we have the vague impression that activation function variants aren't too likely to make a huge different.  But we'd be happy to try something specific out if you point us to it, it would be great to see if there could be improvements made in such a simple way.
>
>
> "CSP was tested with three different curiosity and novelty metrics, none of which dominated over all the other ones. However, PPO was only tested with one of the measures and A2C had no curiosity measure added to it. Where there any preliminary results that justified this decision? In terms of computation and time, how difficult would it be to include this in the paper?"
>
> Fair point.  The reason for what we did is this: once we found PPO to be better than A2C in our context and that RND was if not always the best, at least overall the best curiosity metric, we figured that PPO+RND was thus the overall strongest control of this type. And that very likely the other combinations would just be less powerful.  However, it is easy to include several other combinations, and we will be happy to do this in the revised paper.
>
> "The paper already hinted at this, but macro-actions could be framed within the option framework from Sutton, Precup, & Singh (1999). .... Could the authors provide more comments about this line of future work?"
>
> Yeah, this definitely seems like a natural direction for further improvement.  The key goal is really to get rid of having to build in macro actions. It seems like CSP might be a natural way to do that, and plugging it into the options framework is one clear way forward toward that.  The overall concept of hierarchical planning is something would generally be relevant for us to try to connect to, and it seems natural to think that a hierarchical, and possibly curricularized, version of CSP could be substantially more powerful that the current "single-level" version.

---

> > ### Author Response · Authors · 2019-11-07
> > **Rev 3. follow up on TAMP comparison**
> >
> > Following up in more detail now for this question from Rev3:
> >
> > "... There are no comparisons to any previously proposed Task and Motion Planning Methods. What motivated this decision?"
> >
> > The answer is essentially the same as we gave to Rev 2, who is actually in a way asking a similar question.  The basic answer is TAMP methods basically all requir by-hand specification of the logical definition for action effects and preconditions.   This is the essential reason why TAMP is not that flexible: creating such specifications is a rather laborious and situation-specific process.  You can think of the main raison d'etre of CSP as to get around this problem, using learning (while still retaining the overall efficiency properties of TAMP-based, rather than Deep-RL based, solutions).
> >
> > The problem we'd face if we wanted to implement a TAMP comparisons is: how would we do so without having to make situation-specific logical action effect and precondition specifications?  If we knew how to do that within the traditional TAMP framework, of course we'd do that as an important baseline -- but then that would have solved the problem that CSP is for in the first place!
> >
> > In fact, this problem is essentially the same reason why people in the TAMP community itself don't really do quantitative comparisons between systems in their papers. This is kinda strange sounding for AI/ML community, but is (unfortunately) natural once you realize what the flexibility limitations of TAMP actually are.
> >
> > As we were learning about this field, we found a good youtube video we found which contextualizes this issue: https://youtu.be/wRZ2yqRrPiY?t=4342

---

> > > ### Comment · AnonReviewer3 · 2019-11-13
> > > **Decision after reading all the reviews and comment**
> > >
> > > Thank you for your thorough replies and for addressing our concerns. I am satisfied with the your rebuttal and very please about how receptive you were about all of our feedback. I'd be happy to change my score to an accept as soon as I've verified that the new version of the paper in fact addresses our concerns about reproducibility.
> > >
> > > I would like to add that it seems that the motivation of the paper was lost in all of us, so this could be an area where the paper could be greatly improved.

---

> > > > ### Author Response · Authors · 2019-11-13
> > > > **quick follow up**
> > > >
> > > > Thanks for the reply!
> > > >
> > > > We'll work to improve the clarify of the discussion in the paper about the motivation.
> > > >
> > > > Also, if you find anything missing in the supplement  that you feel is important, just let us know -- we're happy to add / clarify as needed to make what we've done transparent.

---

### Official Review · AnonReviewer2 · 2019-10-23
**Official Blind Review #2**

**Rating:** 1

**Review:**

The idea of the paper is to augment a planner with a curiosity module to reduce the number of traversed paths, resulting in a speedup. The paper presents experiments where it is shown that deep RL methods are outperformed in that question.

I recommend to reject the paper.

The reason for this are threefold.
- The paper is crowded with text and ends up to be hard to follow. The actual contribution is hard to distill.
- The method compares to deep RL baselines. But these are *not* planning algorithms, instead these are RL methods. The paper does not compare to planners, which are tailored to solve the problem the paper adresses.
- The reader is left alone to place the work within the literature. The abstract and introduction do not have a single cite; terms like TAMP and multi-step planning are mentioned and certain properties of them are stated without resorting to where a reader could look those up. It is not the readers task to use a search engine to reverse engineer the authors writing!

I think that the authors need to reformulate what the contribution of their work is. That needs to be presented in a more abstract way, without focusing on the experimental setting prematurely. The experimental setup is ok if the method is restricted to robotic tasks, but too thin for the general setting of efficient planning with sparse costs. Planning algorithms that are tailored towards huge spaces shou;d be used as base lines instead of deep RL methods.

**Experience Assessment:**

I have read many papers in this area.

**Review Assessment: Checking Correctness Of Derivations And Theory:**

I assessed the sensibility of the derivations and theory.

**Review Assessment: Checking Correctness Of Experiments:**

I assessed the sensibility of the experiments.

**Review Assessment: Thoroughness In Paper Reading:**

I read the paper at least twice and used my best judgement in assessing the paper.

---

> ### Author Response · Authors · 2019-11-05
> **What are your suggestions for planning algorithms for us to compare to?**
>
> Hi Reviewer 2!
>
>
> We will get into details of replying to all your comments very soon.  But, one thing we’re hoping to get your thoughts on as soon as possible is about the comment in the last paragraph of your review.
> You say: “Planning algorithms that are tailored towards huge spaces should be used as base lines instead of deep RL methods.”
>
> Could you make a suggestion of what specific comparisons you think we should be using as baselines?
>
> Here’s the main cause of uncertainty we have in addressing your comment:  Are you asking us to compare to standard TAMP solutions?  There’s a fundamental reason we didn’t do this.  We didn’t end up making any comparisons with TAMP algorithms since all the ones we know about require the user to pre-define macro-action effects, preconditions, and predicates.  This makes them quite challenging to apply in the context of the highly flexible setup we are working in.  That's because to figure out the effects and preconditions in a way that can be characterized with formal logic (a basic requirement of TAMP), you kind of have to do it laboriously and specifically for each new robotic setup and macro-action set, and tailor it to the specific objects / structures in the environment.
>
> This lack of flexibility in TAMP is the whole raison d’etre of our paper.  (That’s why we call it “flexible”.)  If it were easily possible to apply TAMP directly as a baseline comparison, that would have obviated the point of our work. It was essentially that we couldn’t really flexibly do this that inspired our work.  We’re definitely open to suggestions about how to make this direct comparison, but just not sure how to make sense of it at the moment, given the limitations of existing TAMP solutions that we know about.
>
> Maybe part of the problem here is that we weren't successful in communicating to you our main contribution.  The main technical problem motivating our work is that standard TAMP (and other planning) solutions are efficient, but hard to apply easily in a flexible context, because they typically require the pre-specification of conditions and actions.  You can view our main contribution as using a specific targeted approach to intrinsically-motived learning to find an efficient but still flexible route around this problem.  Our work shows how one can *still* benefit from the advantages of ideas from planning to achieve efficiency (in a way that standard Deep RL does not).
>
> An alternative possibility: are you referring to other planning solutions like RRT* or KPIECE that don't require the pre-definition of action effects and preconditions? There are many motion planning algorithms like these, but these algorithms have been shown to utterly fail in real-world environments which have many complex differential constraints.  (This is why most TAMP papers don't even compare to pure motion planning anymore. So we didn't think rehashing this well-known point was super-helpful.)
>
> Bottom line though is that we’d be very happy try something out as a new baseline if you make a specific suggestion, especially if the system you suggest has available code we could run in our setting -- but we figure we’d better ask right now to make sure we have the time to do this.
>
> Or if what you're saying is we need to explain better the thoughts above in the paper, we'd be happy to do that.  What do you think?

---

> > ### Public Comment · ~Caelan_Reed_Garrett1 · 2019-11-06
> > **Additional Task and Motion Planning References**
> >
> > There are many planning algorithms for multi-step manipulation problems that do not operate on a discrete abstraction of the domain (what you mean when you say action models with preconditions & effects). Instead, these algorithms search directly in the hybrid state space composed of discrete and continuous variables.
> >
> > The approach of Toussaint, M. et al. is particularly relevant as they also consider tool-use problems. The code from their paper is publicly available: https://github.com/MarcToussaint/18-RSS-PhysicalManipulation.
> >
> > From a satisficing planning point-of-view, the ultimate metric we would like to minimize is runtime; however, your paper only reports the number of samples required. Many existing TAMP algorithms can solve similar problem instances in only a few minutes. How long does planning typically take for the problems you consider?
> >
> > --------------------------------------------------
> >
> > Hauser, K. and Ng-Thow-Hing, V. (2011) “Randomized multi-modal motion planning for a humanoid robot manipulation task,” International Journal of Robotics Research (IJRR). Springer, 30(6), pp. 676–698. Available at: http://journals.sagepub.com/doi/abs/10.1177/0278364910386985.
> >
> > Barry, J., Kaelbling, L. P. and Lozano-Pérez, T. (2013) “A hierarchical approach to manipulation with diverse actions,” in Robotics and Automation (ICRA), 2013 IEEE International Conference on, pp. 1799–1806. Available at: http://citeseerx.ist.psu.edu/viewdoc/summary?doi=10.1.1.365.1060.
> >
> > Vega-Brown, W. and Roy, N. (2016) “Asymptotically optimal planning under piecewise-analytic constraints,” in Workshop on the Algorithmic Foundations of Robotics (WAFR). Available at: http://www.wafr.org/papers/WAFR_2016_paper_11.pdf.
> >
> > Garrett, C. R., Lozano-Pérez, T. and Kaelbling, L. P. L. P. (2017) “Sampling-based methods for factored task and motion planning,” in The International Journal of Robotics Research. doi: 10.1177/0278364918802962.
> >
> > Toussaint, M. et al. (2018) “Differentiable physics and stable modes for tool-use and manipulation planning,” Proc. of Robotics Science & Systems.

---

> > > ### Author Response · Authors · 2019-11-07
> > > **following up on literature comparisons / suggestions and clarifying our core contribution**
> > >
> > > Thanks for the constructive comment!   See below for inline comments.
> > >
> > > "There are many planning algorithms for multi-step manipulation problems that do not operate on a discrete abstraction of the domain (what you mean when you say action models with preconditions & effects)."
> > >
> > > Actually there might be a misunderstanding here.  There's nothing specific about discrete vs continuous here, and working in a discrete setting is not at all what we mean by issue of "logically specified preconditions and effects".  Perhaps you think we are claiming that no continuous action and state space planners exist that solve problems similar to ours. That's not what we are saying. There are obviously many such planners. Instead, what we *are* claiming that while some planners work in continuous action and state spaces, they are made possible and efficient using logical predicates as well as action effects and preconditions that limit the flexibility of the system. Our system forgoes these explicit logical definitions and makes the task possible and efficient via another means, namely curiosity.
> > >
> > >
> > > "The approach of Toussaint, M. et al. is particularly relevant as they also consider tool-use problems. The code from their paper is publicly available: https://github.com/MarcToussaint/18-RSS-PhysicalManipulation."
> > >
> > > We're very familiar with that work. That paper is great and is a nice advice.  However, it isn't an example of something that already resolves the main issue CSP solves. Like all TAMP work we're familiar with, the Toussaint approach requires the logical specification of action conditions and effects.  Specifically, the authors say that:
> > >
> > >    "[We] restrict the solutions to a sequence of modes; consider these as action primitives and explicitly describe the kinematic and dynamic constraints of such modes. This drastically reduces the frequency of contact switches or kinematic switches to search over, and thereby the depth of the logic search problem. It also introduces a symbolic action level following the standard TAMP approach, but grounds these actions to be modes w.r.t. the fundamental underlying hybrid dynamics of contacts.”
> > >
> > > It's also really easy to see from their code how this restriction arises.  For example:
> > >
> > > Action Effects/Preconditions: https://github.com/MarcToussaint/18-RSS-PhysicalManipulation/blob/master/demo/fol.g#L89
> > >
> > > Predictions: https://github.com/MarcToussaint/18-RSS-PhysicalManipulation/blob/master/demo/fol.g#L210
> > >
> > > You can see how in those places they require the definition of operators effects using standard logical form needed for TAMP.  It's exactly this sort of requirement that makes TAMP hard to apply flexibly. Applying the Toussaint algorithm for us would require specific by-hand tuning. That would obviate the whole point of "flexible" planning in the first place.
> > >
> > > The key difference with CSP is that unlike TAMP (and the Toussaint work), we *do not* use specially crafted problem-specific constraints to guide our search, but instead reduce the size of the search tree by using novelty as a guiding heuristic.  This is way CSP is so much more flexible.
> > >
> > >
> > > You also mention some additional literature:
> > >
> > > --> Garrett, C. R. et al. (2017).
> > >
> > > This work falls into the same category as the Toussaint work because it encodes the preconditions and effects of actions into it’s problem-specific “constraint network”.
> > >
> > > --> Barry, J et al. (2013)
> > >    and
> > > --> Hauser, K. et al. (2011)
> > >
> > > These works are solutions for multi-modal motion planning problems which is not the problem we are attempting to solve.
> > >
> > >
> > > --> Vega-Brown, W. (2016)
> > >
> > > This work *IS* actually trying to remove the explicit logical definition from planning and they provide code, and as such we're familiar with it.  But it's really for a very different purpose that CSP.  First, it operates on motion primitives, so it's more like motion planning algorithm with a few non-analytic differential constraints. Second, it was tested on a single fairly simple 2D task with very limited action and state spaces, and it took up to 3 hours to solve that task. For these reasons, the approach is very likely fail to solve any of the problems (even the three-stack task).  To be fair, of course, the authors of that work didn't claim that their work was a solution to the kind of complex multi-step planning problem we address.
> > >
> > > Given the complexity of the algorithm, it would likely be very difficult for us to get a working implementation during this review process.  (They provide Python code but only for a very different 2D setting, so adapting it would be a very substantial effort.)  However, it might be possible.  But also given how unlikely it is to actually even solve the simplest of our multi-step tasks, it doesn't seem obvious this would be effort well spent.
> > >
> > > ==> Question: What do you think?

---

> > > > ### Author Response · Authors · 2019-11-07
> > > > **follow up on runtime question**
> > > >
> > > > "... the ultimate metric we would like to minimize is runtime; .... How long does planning typically take for the problems you consider?"
> > > >
> > > > The time it takes to solve a problem is dependent on many factors such as simulator speed, hardware choices, and the difficulty of the problems being solved. As such, comparing timing between projects which vary on all three seems rather difficult. CSP solves some of the simpler problems in a matter of minutes and with others it takes much longer. We totally agree that runtime is a useful metric, but really will be hard to compare to other things on this unless we're using similar setups and solving similar problems. Once we start applying CSP in the real world, rather than just in simulation, this becomes an important metric, one we hope to report in future work.

---

> > ### Comment · AnonReviewer2 · 2019-11-15
> > **RL solves a different problems; I don't know what a good base line would be, but RL methods are not**
> >
> > Maybe my view on the matter is too much that of an outsider. But still, it appears that dynamic programming (and appropriate approximations from the literature) should be applicable. Maybe the TAMP community has deemed these approaches insufficient, but the targeted community (i.e. not robot motion planning but representation learning) is probably not aware of this–or at least I and many of my peers are not.
> >
> > Instead, you compare to RL methods which are designed to do different things. More concretely, they are designed to work in situations where the dynamics and cost function are *not* given explicitly. Hence, these baselines appear as straw men.
> >
> > How does an approach straight from a text book fail in these settings? Where can I learn about these things? If relevant articles are not cited (I might have overlooked them), I have now way of evaluating your claims. Claiming that these results are "well known" is certainly not true for the ICLR community. It is also something that is not part of all recent textbooks. I don't think you can expect all your readers to be aware of this, or you have to target a different community.

---

### Official Review · AnonReviewer1 · 2019-10-24
**Official Blind Review #1**

**Rating:** 6

**Review:**

This paper tackles the problem of enabling robots to learn long-horizon, sparse-reward tasks. The proposed approach, the Curious Sample Planner (CSP), builds on insights in task and motion planning (TAMP), which is a standard approach for tackling these kinds of tasks. TAMP constructs a plan in the space of macro-actions (e.g., move object 1 to location (x,y)), and uses a motion planner to execute each macro-action. However, TAMP typically requires being able to describe macro-action effects and preconditions with logical predicates, which can be impossible in real-world environments, due to complex dynamics and interactions. CSP overcomes this limitation by planning in the space of macro-actions in a way that is biased toward novelty.

The core approach of CSP is to train a (macro-)action selection network to generate macro-actions that are both feasible and novel. The curiosity module is used in two ways: (1) to give reward to the action selection network for producing novel macro-actions, and (2) to expand states that are considered novel. Three state-of-the-art ways of computing novelty are compared -- state estimation (SE), forward dynamics (FD), and random network distillation (RND).

CSP is evaluated on a suite of simulated robotics tasks that require the robot to build simple machines from the objects in its environment, in order to achieve the specified objective. The experiments compare agents trained with CSP versus with deep RL (specifically, A2C, PPO, and PPO + RND). There is an ablation study, that compares against planning with uniform selection of macro-actions and uniform selection of states to expand.

Overall, this paper makes a significant contribution to improving robot learning of long-horizon, sparse reward tasks. The paper is clearly written and well-motivated, and the evaluations are thorough. The major downside is that CSP inherently requires knowing the dynamics of the environment (i.e., having a simulator), which means it cannot be directly run applied to real-world robotic systems. But it is a step in the right direction, and clearly outperforms vanilla deep RL.

I'm leaning toward accept, but I have a few concerns / questions about the paper. First, there are not enough details included for reproducibility (see list below). With regard to evaluation, I think another ablation should be run, where the action selection network is trained for only feasibility. This would be similar to CSP-No Curiosity, but with a reward of 0 for infeasible actions, and a small fixed positive reward for feasible ones. This would more clearly answer the question of how much it matters to include the curiosity module. Finally, I'm not convinced that this approach works well for transfer, and the evaluations seem inconclusive as well. I'm surprised that even for inter-task transfer, agents trained with action selection transfer and full transfer don't just learn to solve the task immediately. Am I missing something here about how the task is instantiated?

Reproducibility questions:
- In Algorithm 1, what are the inputs to the novelty metric that is used to compute L_\phi? Is it the batch of next states, S'?
- What is the form of the output of the action-selection network? And what exactly is the space of macro-actions? For instance, the number of possible RemoveConstraint macro-actions depends on how many objects are connected in the environment. But the dimension of the action-selection network's output must be fixed.
- What does the state vector input for FD and RND contain? Along these lines, why not also use image inputs for FD and RND, as is done for SE?
- What are the learning hyperparamters used to train the networks? (e.g., learning rate)
- How many perspectives (i.e., n_p) are used for the SE curiosity module?

Minor comments / typos:
- Avoid using the same variable with different meanings, e.g. using \phi to indicate both the curiosity module and the parameters of the value network.
- Page 3: "flexible", "a flexible"
- Page 5: "learnabe" --> "learnable"
- Page 8: "in which" --> "in which the"
- Page 9: "illustrate" --> "illustrated"

**Experience Assessment:**

I have published one or two papers in this area.

**Review Assessment: Checking Correctness Of Derivations And Theory:**

I carefully checked the derivations and theory.

**Review Assessment: Checking Correctness Of Experiments:**

I carefully checked the experiments.

**Review Assessment: Thoroughness In Paper Reading:**

I read the paper thoroughly.

---

> ### Author Response · Authors · 2019-11-07
> **First response to Review 1**
>
> Hi Reviewer 1!
>
> Thanks for your constructive feedback.  See below for inline replies to each major item.
>
> ...
>
> "Overall, this paper makes a significant contribution to improving robot learning of long-horizon, sparse reward tasks. The paper is clearly written and well-motivated, and the evaluations are thorough.... It is a step in the right direction, and clearly outperforms vanilla deep RL."
>
> Thanks!
>
> "... The major downside is that CSP inherently requires knowing the dynamics of the environment (i.e., having a simulator), which means it cannot be directly run applied to real-world robotic systems."
>
> Absolutely.   As you probably realized, in this world we've taken a step-by-step strategy: first try to get the planning working *assuming* there is good forward predictor, so that in the next phase of the work we can relax that condition. Our next major step is to apply the method in the context of a non-deterministic learned forward predictor (and in fact we hope to show that having a good planning algorithm in place makes the learning of the forward predictor substantially more efficient).  As you note, doing this is really important for actually being able to apply to our work in the real world.
>
>
> "... First, there are not enough details included for reproducibility (see list below)."
>
> Yes, both you and Reviewer 3 had this concern, and its totally reasonable.  We'll reply to the details of your concerns below, but here are the three main high-level things we're going to do in response:
>
> 	1. Revise the main text of the paper clarifying the several main important issues for which you and Rev 3 have asked.
> 	2. Create an additional supplementary document with copious detailed information about each aspect of the project.
> 	3. Make a public github account/repo to which we will commit all the project code, so that you (and others) can have access to it during (and after) this review process. We should have done this before but somehow got worried about breaking the double-blindness of the review. However, upon thinking about it, we realize it should be easy enough to create an anonymous Github user account to post the code to for review purposes.
>
> Our plan is to have this ready for your review by 11/11 or 11/12.  That will give you a few days to ask for further clarifications.
>
> ==> Question: does this plan work for you? Do you think it's enough time to address the issues?
>
>
> "... With regard to evaluation, I think another ablation should be run, where the action selection network is trained for only feasibility. This would be similar to CSP-No Curiosity, but with a reward of 0 for infeasible actions, and a small fixed positive reward for feasible ones. This would more clearly answer the question of how much it matters to include the curiosity module."
>
> Just a point of clarification, removing the curiosity feedback into the action selection networks is not the same as CSP-No Curiosity. In fact, most of the benefit of curiosity comes from selecting which nodes to add to the search tree and the frequency with which to sample those nodes. This operation is independent of the action-selection networks. A more informative ablation might be to run CSP without feasibility feedback. Comparing this result with CSP-No Curiosity would isolate the contribution of the curiosity feedback signal independent of feasibility. Is this an ablation you would be interested in seeing?
>
>
> "... Finally, I'm not convinced that this approach works well for transfer, and the evaluations seem inconclusive as well. I'm surprised that even for inter-task transfer, agents trained with action selection transfer and full transfer don't just learn to solve the task immediately. Am I missing something here about how the task is instantiated?"
>
> Transfer is tested on different initial instantiations of the general problem statement. So for example in 3-stack, the blocks are placed in different starting positions. It's important to note that we are not training a policy to solve these tasks, since a valid and general policy may not be learnable from a single example. The reason for showing between-task transfer was only to show that there is some increase in efficiency from having solved the tasks before. I'm also curious to know how much the log scaling on the y axis contributed to your interpretation of the results as being inconclusive. (In some cases the efficient gain was substantial even when the log-scale made it look small.)
>
> "Reproducibility questions: [LIST OF SPECIFICS]"
>
> Ok, we'll make sure to add information about all these things in either the revised main text or the supplement.  Once we post these, we'll write another comment pointing you to where the answers have been added.
>
>
> "Minor comments / typos:... [LIST OF SPECIFICS]"
>
> Thanks for catching these.  All typos have now been fixed and will be uploaded along with other revisions.

---

### Author Response · Authors · 2019-11-13
**Updated Revision**

General comment:
*To address issues regarding reproducibility and implementation details, we have provided open-sourced all the code behind our project at the following anonymized Github repo: https://github.com/CuriousSamplePlanner/CuriousSamplePlanner

Reviewer 1:
* We have addressed questions of the novelty metric used in the generation of L_\phi in the supplement where we review the details and loss function of each curiosity type.
* We have addressed questions about the action space in section C.6 of the supplement where we describe the exact composition of the action space used in training the curiosity networks and action selection networks.
* We have added details about the exact input and output of each curiosity metric (Forward dynamics, state estimation, random network distillation) in section A of the supplement.
* We have added information regarding network structure in section B of the supplement and learning rate/additional hyperparameters in section  D of the supplement.
* We have addressed the number of perspectives (one) used in calculating the state-estimation curiosity heuristic in section A.1
* We have run an additional ablation on the action selection network and reported our results in the main body of the paper. R1 requested that we run an ablation on the feasibility aspect of the action selection networks. We decided to go even further and run an ablation on the entire action selection network. The reason we did this is that in originally designing our architecture, we started without having the action selection network. This seemed like it would be sufficient for solving individual tasks. But we felt that having it would be useful for transfer to new tasks since it would enable task-independent learning.  Our new ablation shows that, while the action selection networks can be removed without dramatically disrupting the initial planning process, it is indeed vital for task transfer.
* We fixed all five of the mentioned typos in the main body of the paper

Reviewer 2
* We followed up with reviewer 2 about the key contribution of our paper: we have built a TAMP-like multi-step planning algorithm that uses deep-learning-based curiosity to enable flexible application, as opposed to having to explicitly specify action effects/preconditions and logical predicates as in traditional TAMP
* We addressed our absence of comparisons to task and motion planning algorithms in followup comments with reviewer 2.
* Following the advice of reviewer 2, we have added a few citations to the introduction and included some of the mentioned research in the related work section.

Reviewer 3
* We have added all needed implementation details, including the hyperparameters used for A2C and PPO implementation in the supplement of the paper.
* We have discussed in responses to reviewers 2 and 3 why it wasn’t obviously feasible to make a direct quantitative comparison to traditional task and motion planning algorithms.
* We discussed with reviewer 3 our choice of actor-critic architecture, activation functions, and comparisons.
* We performed some additional baseline comparisons with A2C/PPO on the other curiosity metrics, finding similar and expected results (section E of supplement and referenced in the main text).
* We discussed macro-action/option discovery and what form this may take in future work.

---

> ### Comment · AnonReviewer3 · 2019-11-14
> **Further clarification**
>
> Thanks again for all the effort you are putting into addressing our comments. The submission is already a lot stronger than the original one. However, I still have a few more concerns.
>
> First, could you provide some details about how the hyperparameters were selected? It seems unreasonable to me to use a value of zero for gamma in Vanilla PPO and Vanilla A2C because this would make the algorithms very myopic in an environment with very sparse rewards. Thus, it seems hopeless that either of these two algorithms would succeed in most of this tasks, which explains their performance. If this was the only value of gamma used for Vanilla PPO and Vanilla A2C, then I don't think this is a fair comparison.
>
> Second, could plots and tables be provided for the performance of the other baselines algorithms in appendix E?
>
> Finally, the organization in the appendices is slightly odd. I think the information would be easier to follow each method section in the appendix included all the details about the method, i.e., hyperparameters, loss, and architecture. Moreover, Appendix D.1 seems to end mid sentence.

---

> > ### Author Response · Authors · 2019-11-14
> > **Hyperparameter Choice Clarification**
> >
> > You are correct that in most reinforcement learning problems with sparse reward, gamma is usually between 0.9 and 0.99. However, we chose gamma to be zero for a very specific reason.
> >
> > There are essentially two cases: those in which we use curiosity and those not (the "vanilla" cases).
> >
> > In the case with curiosity, although the original goal is very sparse in the state space, the reward given to the reinforcement learning algorithm becomes *not  at all* sparse, since it is the loss from the curiosity module. The intrinsic motivation has filled in what *was* a sparse reward into something quite dense.  This loss creates a reward gradient in the direction of increasing uncertainty. While we started our experimentation with the default gamma of 0.95, we actually found higher performance in CSP transfer for gamma=0, and no change in single-task performance.
> >
> > In the cases of Vanilla PPO and A2C, where there is no intrinsic motivation and the reward is truly very sparse, something totally different (and almost opposite!) is going on to make gamma=0 "reasonable".  If you think about what gamma actually does, it only matters at all if the RL agent receives *at least one positive sample of achieving reward*.  But because of the incredibly low probability of finding a reward at all through non-directly sampling at the beginning of the RL learning process, the vanilla agents *never* even get one instance of reward througout the whole episode.   So gamma ends up being totally irrelevant.  If we had waited billions and billions of episodes, perhaps for some of the tasks the agent woul have received some reward, and then of course, gamma would have started to matter.   But that would have defeated the point of the work to begin with (e.g. *not* needing billions of steps to figure these problems out). This is kind of a restatement of the underlying reason why for multi-step  deep reinforcement learning, even when trained on auxiliary inputs, is incapable of solving long-horizon multi-step planning tasks with very sparse goals.
> >
> > Sorry, we realize in retrospect that this is fairly subtle and requires explaining.  We have added these clarifications to section B.1 of the paper.
> >
> > Regarding your other suggestions, we have added a table in the supplement with the additional experiments, fixed the typo that you found, and restructured the supplement, removing section D and transferring all of the hyperparameters in that section into their respective architecture sections.
> > The supplement now contains 4 sections which describe curiosity types, architectures, experiment details, and additional baselines. If there’s anything about the structure that you feel is still confusing, please let us know.
> >
> > Thank you again for your helpful feedback.

---

> > > ### Comment · AnonReviewer3 · 2019-11-14
> > > **More about the value of gamma and the task specification**
> > >
> > > I understand that CSP does not require a high value of gamma because it is maximizing the novelty score from the curiosity model. Moreover, since states are sampled from the search tree based on their curiosity score, I can see why using a low value of gamma would not have a big impact on performance.
> > >
> > > I also see how, given the task specification, the value of gamma would not have a huge effect on the performance of Vanilla A2C and Vanilla PPO. After all, the task consists of solving a very complex problem with no feedback about how well you're performing until the task is completed. In RL this type of tasks would take several episodes of training before an agent achieves a reasonable performance. That's why it seems to me that it is hopeless for Vanilla A2C and PPO to succeed at this task. Even when using the curiosity score to guide exploration, curiosity would not necessarily guide the algorithm towards states that are closed to the goal. Thus, I don't think these are the appropriate baselines to compare against.
> > >
> > > I think a good baseline to compare against would be a model-based deep RL method with access to the simulator (the perfect model) and a curiosity module to prioritized states sampled from the model. This would be something akin to the Dyna architecture with prioritized sweeping based on the curiosity score of each state,  see Sutton & Barto (2018) for more information.
> > >
> > > === References ===
> > > Sutton, R. S., Barto, A. G. (2018 ). Reinforcement Learning: An Introduction. The MIT Press.

---

> > > > ### Author Response · Authors · 2019-11-14
> > > > **reply to about proper baselines**
> > > >
> > > > Ach, yeah, you're totally right.  In trying to figure out what the proper baselines where, we went through a thought process exactly like this:  are standard Deep RL algorithms so doomed that it isn't even a fair comparison?  In a way, what we've done with our controls ends up being kinda of a pedagogical exercise making this point.
> > > >
> > > > And as you say, "Even when using the curiosity score to guide exploration, curiosity would not necessarily guide the algorithm towards states that are closed to the goal." -- which we've shown a bit with our PPO-RND (and related) baselines.
> > > >
> > > > So agreed, these baselines are doomed for these tasks essentially, but they do represent  the recent standards for what one might do with deep RL (especially something like PPO-RND).  We probably at least had to make some gesture toward these comparisons.
> > > >
> > > > But a really proper baseline would have to be like what you're suggesting.  It's possible that it might not even really be a baseline, but instead a potentially reasonable alternative model.    It's probably too late for us to do anything about it now (sorry!) especially since this would probably involve fairly nontrivial new implementations and testing, and implementing something like dyna-Q for our situation isn't exactly an off-the-shelf solution, at least not in the high-dimensional continuous setting.  But yeah, we totally agree that's the right direction for exploring strong model comparisons.   We kind of wish we had had this in mind 10 days ago... Maybe something like this can be done for a final version of the paper.

---

> > > > > ### Comment · AnonReviewer3 · 2019-11-14
> > > > > **Useful discussion**
> > > > >
> > > > > I do regret that this came up so late during the reviewing process. However, don't feel discouraged about this. Even if it's too late to address this issue in time for the decision deadline, I think what we have discussed here will be very useful for future submissions. I see the value of your work and I consider it a meaningful contribution, so do continue working on it. Given how receptive you have been about our feedback, I am confident that this work will eventually be published in one of the major conference in our field.
> > > > >
> > > > > In the meantime, I've updated my official rating, but I still consider that the paper is still not ready for publication.

---

> > > > > > ### Author Response · Authors · 2019-11-14
> > > > > > **follow up on Dyna**
> > > > > >
> > > > > > Upon thinking about your suggestion about Dyna further (the co-authors have been going through this extensively over the past few hours), what we’ve realized is that we have built is, at least arguably, a sort of Dyna-like model, but in which:
> > > > > >
> > > > > >   (a) the data-structure of our replay buffer is a tree, and,
> > > > > >
> > > > > >   (b) priority is generated from the learned curiosity-based score.
> > > > > >
> > > > > >   (c) the dynamics are perfect (for now; obviously we want to change this later)
> > > > > >
> > > > > > That is, the baseline you described in your last message may not really be so much an “alternative” separate baseline, but actually a loose re-description of our approach.  Probably whether you’d consider this to be the case or not depends on what you think is “core” to dyna.
> > > > > >
> > > > > > At a higher level, we now realize that one can rephrase our contribution as linking several different research areas together -- that is, TAMP and model-based RL -- by building a model of hybrid kind that shares some features of both.  And we’ve shown how this hybrid approach is very effective for a wide class of hard but very important and unsolved planning problems.
> > > > > >
> > > > > > We just hadn’t been thinking about it exactly in those terms ourselves until now, as we had been more approaching the problem from the point of view of increasing the flexibility of TAMP with learning rather than adding planning to a reinforcement learning problem.
> > > > > >
> > > > > > Do you agree that this is a fair description?
> > > > > >
> > > > > > --> If so, maybe what we can do right now easily is add to the paper by explaining this connection clearly.  In that case, might it not be fair to change score?  And publishing the work close to as-is, since in this case, it’s not really that an important totally different baseline has not be run, but rather that a connection to the literature needed to be explained?
> > > > > >
> > > > > > --> If not, it would be super helpful to us for future work if you’d be able to comment on which specific differences we should be focusing on to test, as distinct baselines and providing citations for where this work has been tested before.

---

### Decision · Program_Chairs · 2019-12-19

**Decision:**

Reject

**Comment:**

The authors consider planning problems with sparse rewards.
They propose an algorithm that performs planning based on an auxiliary reward
given by a curiosity score.
They test they approach on a range of tasks in simulated robotics environments
and compare to model-free baselines.

The reviewers mainly criticize the lack of competitive baselines; it comes as now
surprise that the baselines presented in the paper do not perform well, as they
make use of strictly less information of the problem.
The authors were very active in the rebuttal period, however eventually did not
fully manage to address the points raised by the reviewers.

Although the paper proposes an interesting approach, I think this paper is below
acceptance threshold.
The experimental results lack baselines,
Furthermore, critical details of the algorithm are missing / hard to find.